# A targeted CRISPR-Cas9 mediated F0 screen identifies genes involved in establishment of the enteric nervous system

**Rodrigo Moreno-Campos[1,2], Eileen W. Singleton[1,2], Rosa A. Uribe[1,2] ***

**1** Biosciences Department, Rice University, Houston, Texas, United States of America, **2** Laboratory of Neural Crest and Enteric Nervous System Development, Rice University, Houston, Texas, United States of America

\* rosa.uribe@rice.edu

**Data Availability Statement:** All relevant data are within the manuscript and its Supporting information files.

## Abstract

The vertebrate enteric nervous system (ENS) is a crucial network of enteric neurons and glia resident within the entire gastrointestinal tract (GI). Overseeing essential GI functions such as gut motility and water balance, the ENS serves as a pivotal bidirectional link in the gut-brain axis. During early development, the ENS is primarily derived from enteric neural crest cells (ENCCs). Disruptions to ENCC development, as seen in conditions like Hirschsprung disease (HSCR), lead to the absence of ENS in the GI, particularly in the colon. In this study, using zebrafish, we devised an *in vivo* F0 CRISPR-based screen employing a robust, rapid pipeline integrating single-cell RNA sequencing, CRISPR reverse genetics, and high-content imaging. Our findings unveil various genes, including those encoding opioid receptors, as possible regulators of ENS establishment. In addition, we present evidence that suggests opioid receptor involvement in the neurochemical coding of the larval ENS. In summary, our work presents a novel, efficient CRISPR screen targeting ENS development, facilitating the discovery of previously unknown genes, and increasing knowledge of nervous system construction.

## Introduction

The Enteric Nervous system (ENS) is an extensive and complex network of enteric neurons (EN) and glial cells that inhabit the length of the gastrointestinal tract (GI). The ENS controls inherent GI functions, such as gut motility and intestinal barrier function [1]. In humans, the ENS is estimated to contain over 600 hundred million neurons intrinsically located between the two muscular layers of the GI tract [2]. The ENS has been referred to as "the first brain", as it has been hypothesized as the first evolved nervous system in all extant animal species [3]. Accumulating evidence has revealed ENS importance as a bridge in the microbiota-gut-brain (MGB) axis by establishing bidirectional connections between the Central Nervous System (CNS) and microbiota [4, 5].

For ENS to be functional, it requires the adequate development, differentiation, and assembly of its many different ENs and glial cells during development. In vertebrates, the ENS is

**Funding:** This study was supported by the National Institutes of Health grant R01DK124804 awarded to R.A.U., and by National Science Foundation grant 1942019 awarded to R.A.U. The funders had no role in study design, data collection and analysis, decision to publish, or preparation of the manuscript.

**Competing interests:** The authors have declared no competing interests exist.

primarily derived from neural crest cells (NCC) [6, 7]. NCCs are proliferative, highly migratory, and multipotent stem cells that delaminate and migrate from the length of the embryonic neuraxis based on microenvironmental cues. NCCs migrate extensively throughout the developing embryo and differentiate into a multitude of different cell types, such as craniofacial tissues or pigment cells [8] NCCs commit to an enteric lineage once they enter the foregut mesenchyme, at which point they are named enteric neural crest cells (ENCCs). ENCCs express a combination of marker genes that encode transcription factors and receptors, including *Sox10*, *Foxd3*, *Phox2b*, *Ret* and *Gfra1* [9, 10]. From the foregut, ENCCs continue to migrate caudally into distal hindgut, where they differentiate into ENs, which are classified based on a combination of molecular and cellular means [8–10].

There have been great efforts in studying how the ENS forms during early embryonic development due to the numerous ENS diseases known to afflict children and adults. These diseases include Hirschsprung disease (HSCR), a severe enteric neuropathy marked by absence of ENS in the distal gut, leading to gut dysmotility and/or Megacolon, and affecting 1 in 5000 newborns [11]. In addition to HSCR, defective ENS function can cause Esophageal Achalasia, Chronic Constipation, and Gastroesophageal Reflux Disease, affecting adults and children worldwide [12].

Zebrafish has gained prominence as a pertinent vertebrate model for biomedical and ENS research. Zebrafish generate plentiful externally fertilized eggs, develop transparent embryos, and share 70% genetic sequence similarity with humans, among which 84% correlates with known human-associated diseases [13]. Even though zebrafish ENS is less complex in architecture compared to mammals, lacking one layer of ENs (submucosal plexus) and displaying scattered ENs in contrast with the clustering of ganglia, zebrafish ENS and GI functions are largely conserved with mammals [14, 15]. Notably, various stages of zebrafish ENS development, genes, and signaling pathways have been elucidated, contributing to a deeper understanding of the molecular basis of ENS development [14, 15]. In zebrafish, ENCCs migrate into and along the developing gut between 32–72 hours post fertilization (hpf), and by 4 days post fertilization (dpf) an ENS network begins to form around the whole length of the GI tract [15–19]. Thus, its genetic conservation, and rapid, simple ENS development, make zebrafish an attractive animal model for elucidating ENS development.

Recently, integration of single-cell RNA sequencing (scRNA-seq) into the zebrafish model for exploration of ENS development has yielded invaluable insights into what genes are expressed during ENS developmental phases [20, 21]. Concurrently, the successful adoption of CRISPR technology for gene disruption in zebrafish, known for its high efficiency in analyzing phenotypes directly in the injected generation (F0), also known as "crispants", facilitates swift assessment of candidate genes [22, 23]. This synergy between scRNA-seq and CRISPR technology not only holds promise to enhance our understanding of ENS development, but also allows for the rapid and targeted interrogation of novel candidate genes.

In this study, we employed a targeted F0 CRISPR screen, informed by scRNA-seq data [20] of early zebrafish ENS development. We pinpointed twelve genes, that when disrupted, led to various ENS development phenotypes across different F0s, or crispants. In particular, we discovered that crispant fish targeting genes encoding for opioid receptors, *oprl1* and *oprd1b*, presented with severe ENS development defects, in which further phenotyping showed reduced ENCC numbers resident along the gut. Subsequently, *oprl1* and *oprd1b* crispant larvae displayed alterations in EN neurochemical coding during ENS maturation. Our subsequent focused investigations of the opioid pathway affirmed its pivotal role in ENS establishment along the gut length, whereby temporal opioid pathway inhibition reduced ENCC abundance along the gut.

## Results

### Construction of an *in vivo* ENS-focused F0 CRISPR screen

Leveraging scRNA-seq datasets is an attractive method for uncovering novel genes expressed during ENS development. Candidate genes can then be targeted with CRISPR gene editing to aid us in understanding their potential functional roles during ENS development. To that end, we focused on our prior zebrafish embryo-to-larval stage single-cell atlas that contained *sox10*:GFP-expressing and -derived cells [20] for further analysis. Previously, cell clusters from the 68–70 hpf *sox10*:GFP dataset that captured neuronal populations based on the combinatorial expression of enteric neuron markers such as *elavl3*, *phox2bb*, *ret* and *gfra1a* (S1A Fig), were subset and re-clustered, yielding five sub-cluster populations (0–4) (Fig 1A) [20]. Functional enrichment and interactome analysis [24] identified cellular and signaling pathways related to neurons such as membrane trafficking, neuronal system and axon guidance (S1B–S1D Fig). A total of twelve genes, with high cell-expression distribution from sub-cluster 3 (Fig 1B), and which were associated with various predicted neuronal functions, such as receptors, neuropeptides and transcription factors (S1B–S1D Fig), were selected for reverse genetic analysis to elucidate their functional significance (Table 1). These genes were also present in the protein-protein interaction (PPI) network of sub-cluster 3 in STRING (S1E Fig, S1 Data) [25].

Next, the candidate ENS genes were input into a targeted ENS F0 CRISPR screen (Fig 1C). The screening strategy comprised sequential steps, with each experimental phase requiring confirmation from the preceding one to ensure the integrity and completion of the entire screen. This involved not only genotyping validation and high-content phenotyping but also a meticulous validation process at each juncture to maintain methodological robustness to uncover phenotypic alterations during ENS development, as shown in Fig 1C. Leveraging Tg

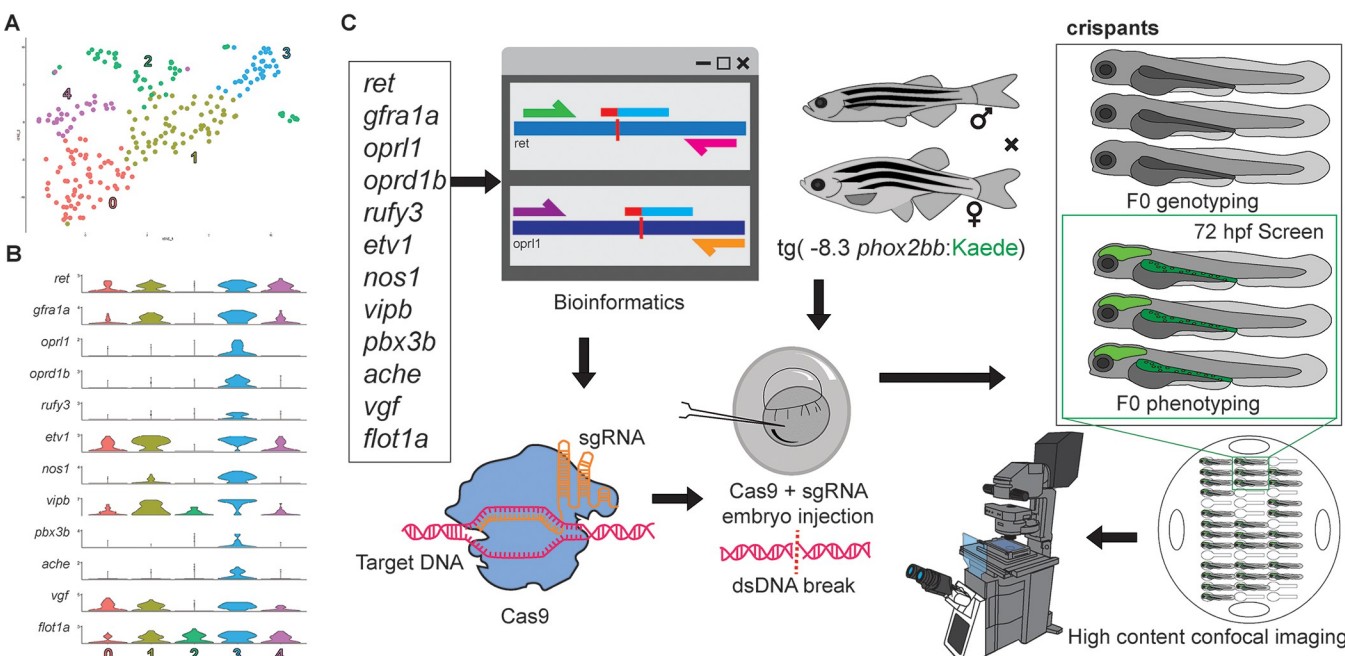

**Fig 1. Construction of an F0 CRISPR screen for ENS development.** (A) tSNE plot shows five distinct sub-clusters after the subset analysis and re-clustering of Clusters 5 and 12 from the [20] 68–70 hpf data set. (B) Violin plots reveal high single-cell expression distribution in sub-cluster 3 of ENS candidate genes. (C) Twelve genes underwent a comprehensive CRISPR screen, involving bioinformatic design, and CRISPR-Cas9 mutagenesis in -8.3*phox2bb*:Kaede zebrafish larvae to visualize enteric cells. The screening strategy included subsequent genotyping validation and high-content phenotyping.

**Table 1. Nomenclature, function, and crispant ENS-associated phenotypes from genes of the CRISPR screen.**

| Zebrafish gene symbol | Zebrafish gene name | Generally known encoded protein functions | Reference | Crispant phenotype |
|---|---|---|---|---|
| ret | ret proto-oncogene receptor tyrosine kinase | Member of the GDNF family ligands and receptor. Receptor tyrosine kinase transmembrane protein. Requires GDNF family ligands (GFLs) and GDNF family receptor alphas (GFRαs) for intracellular signaling activation. | [16, 26] | aganglionosis |
| gfra1a | gdnf family receptor alpha 1a | Member of the GDNF family ligands and receptor. Cell Surface receptor and Co-receptor of Ret, GDNF family ligands (GFLs) bind to them to form GFL-GFRαs-Ret signaling complex. | [27] | hypoganglionosis |
| oprl1 | opiate receptor-like 1 | Transmembrane proteins belonging to the super-family of G protein-coupled receptors (GPCRs). Part of nociception/ orphanin FQ (NOP). In the nervous system endogenous and exogenous opioids exert action through them modulating emotions, memory, neuroprotection and analgesia. | [28] | hypoganglionosis |
| oprd1b | opioid receptor, delta 1b | Transmembrane proteins belonging to the super-family of G-protein-coupled receptors (GPCRs). delta opioid receptor (DOR). In the ENS it modulates the DOR-Enkephalin Axis. | [28, 29] | hypoganglionosis |
| rufy3 | RUN and FYVE domain-containing 3 | Endolysosomal protein that promotes coupling of endolysosomes along microtubules. Has been implicated in regulating neuronal polarity and axonal growth. | [30, 31] | hypoganglionosis |
| etv1 | ETS variant transcription factor 1 | Transcription factor member of the ETS (E twenty-six) family. Plays a role in orchestrating the neural activity-dependent gene regulation for terminal maturation of brain granule neurons and part of the transcription factor combinational codes during the differentiation of ENCCs branches. | [32, 33] | hypoganglionosis |
| nos1 | nitric oxide synthase 1 (neuronal) | Synthesizes nitric oxide in the ENS playing and important role in synaptic transmission, muscular tone, mucosal barrier function and fluid secretion. | [34] | normal |
| vipb | vasoactive intestinal peptide b | Neuropeptide released by VIP-producing neurons, in the ENS regulates microbiota and mucosal barrier homeostasis. | [35, 36] | hypoganglionosis |
| pbx3b | pre-B-cell leukemia homeobox 3b | PBX3 belongs to the conserved PBX family of TALE (3-amino acid loop extension) homeodomain transcription factors. Regulates the transition of postmitotic inhibitory to excitatory neurons in the ENS. | [33] | normal |
| ache | acetylcholinesterase | AchE, an enzyme that catalyzes the breakdown of the neurotransmitter acetylcholine. Secreted by the excitatory ENs stimulating muscle contractions, intestinal secretions, release of hormones and blood vessels dilation. Absence of ganglia is associated with increase in AChE. | [37, 38] | hypoganglionosis |
| vgf | VGF nerve growth factor inducible | Neuroendocrine regulatory polypeptide. Part of the Nerve growth factors (NGF). Distributed in neurons and neuroendocrine tissues. Derived polypeptides regulate energy, water balance, circadian rhythm. Associated with depression, Alzheimer's disease and other neuroendocrine diseases. | [39] | hypoganglionosis |
| flot1a | flotillin 1a | Part of the protein family that includes a Stomatin Prohibitin Flotillin HflK/C (SPFH) domain. Part of lipid rafts. Participates in clathrin endocytosis, signal transduction, extracellular vesicles, and membrane trafficking. Important for the development of the hippocampal neurons and mediates excitatory synaptic transmission in the brain. | [40] | hypoganglionosis |

(-8.3phox2bb:Kaede) transgenic zebrafish to identify ENCCs and ENs fluorescently [18], we implemented our research strategy by injecting single guide RNA (sgRNA) in complex with Cas9 protein into 1-cell stage transgenic embryos targeting each specific gene (S1 Table), and phenotyping at 72 hpf when the ENS is undergoing neurogenesis. Each experimental set of injected CRISPR F0 embryos, "crispants", were subjected to genotyping validation and phenotype determination, as outlined in Fig 1C, and described below.

## Candidate gene crispants have ENS genotypic and phenotypic alterations

Genotyping subsets of the crispants for each specific targeted gene via T7 endonuclease 1 (T7E1) mismatch assay was used to detect indel presence, and to enable downstream phenotyping assays (Fig 2A) for each batch of F0s. Specifically, our experiments detected indels in a

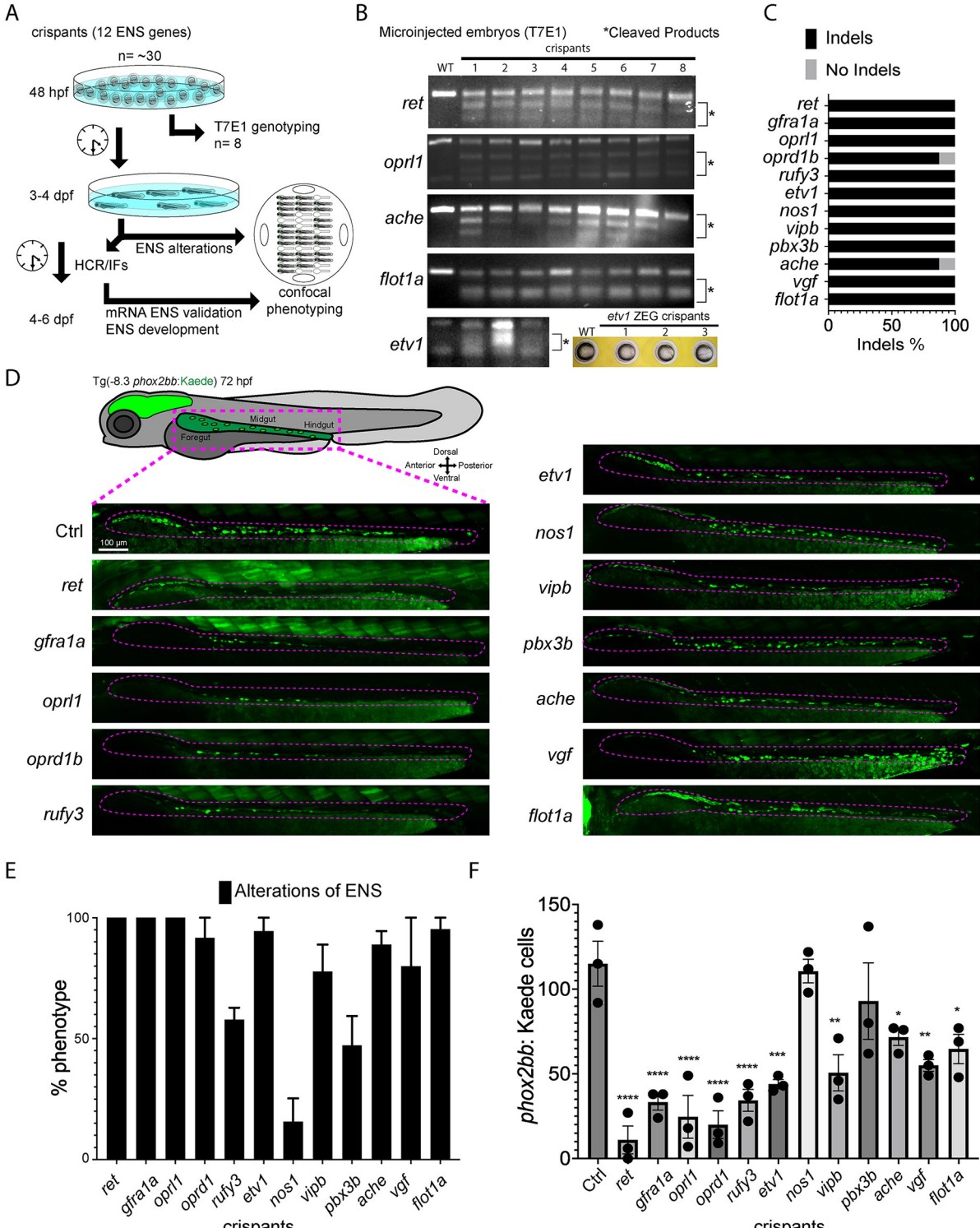

**Fig 2. Candidate genes targeted in an F0 CRISPR screen display ENS phenotypic alterations.** (A) At 48 hpf, eight crispant embryos from pools of around thirty embryos were used to validate CRISPR activity via T7E1, for each gene targeted. If the majority had indels, then subsets of the pool were grown at 3 to 4 dpf to phenotype their ENS. The phenotyping process combined crispants of different genes by using an agarose cast that enabled high-content semi-automated confocal imaging. An additional fraction of the crispants were analyzed at 4–6 dpf for additional HCR validation or for late phenotypic alterations. (B) Representative images of different T7E1 assays demonstrate indels of different embryos in the specific gene-targeted regions. *etv1* gene was genotyped using ZEG. The asterisks denote the presence of cleaved

products. (C) Percentage of embryos with CRISPR/Cas9 induced indels of different ENS genes ($\geq 2$ experiments). (D) Confocal images of Tg (-8.3$phox2bb$:Kaede) ENCCs/ENs for different crispants along the gut at 72 hpf. ENCCs/ENs of most crispants failed to localize distal hindgut ($\geq 2$ experiments). (E) Pools of the twelve ENS gene crispants showing the percentage of phenotypic alterations ($\geq 3$ experiments). (F) Number of fluorescent ENCCs and/or ENs along the gut from the different gene crispants ($\geq 3$ experiments with 3 biological replicates). Comparing the mean of the control with the mean of each gene, ANOVA P value ****: <0.0001, ***: 0.0004, **: <0.003, *: <0.05.

high percentage of embryos (Fig 2B and 2C). This validation assured us that the remaining crispants from each injection pool could be examined for downstream phenotypic ENS alterations. To further support that the imaged embryos during confocal imaging had mutagenesis, subsets of the injected embryos were genotyped live before imaging using the Zebrafish Embryonic Genotyper (ZEG, Fig 2B).

To identify ENS phenotypic alterations in the -8.3$phox2bb$:Kaede crispants, we first qualitatively performed our CRISPR screen by examining the colonization success of ENCCs/ENs along the developing gut at 72 hpf. As positive controls for this screen, we utilized sgRNAs against the tyrosine kinase receptor gene, $ret$ (REarranged during Transfection), and the GDNF family receptor alpha-1 gene, $gfra1a$ (Fig 2D). As previously reported, $ret$ and $gfra1a$ loss-of-function larvae display aganglionosis and hypoganglionosis phenotypes, respectively [16, 17, 27, 41], with aganglionosis presenting as near complete loss of ENs, while hypoganglionosis presents with reductions. Hypoganglionosis alterations were identified in crispants from eight of the screened candidate genes. These genes included: the opioid receptor encoding genes $oprl1$ and $oprd1b;$ RUN and FYVE domain containing 3 protein-encoding gene, $rufy3$; ETS (E twenty-six) variant transcription factor 1 encoding gene, $etv1$; Vasoactive intestinal peptide b encoding gene, $vipb$; Acetylcholinesterase encoding gene, $ache$; VGF nerve growth factor inducible encoding gene, $vgf$; and the membrane-associated protein Flotillin 1 encoding gene, $flot1a$. The Nitric oxide synthase encoding gene, $nos1$, and Pre-B-cell leukemia transcription factor 3b encoding gene, $pbx3b$, didn't show overt phenotypic alterations in colonization. The percentage of crispants for each gene with ENS phenotypic alterations was over 80% with all genes except for $rufy3$, $nos1$, and $pbx3b$ (Fig 2E). Further phenotypic examination of -8.3$phox2bb$:Kaede[+] larvae crispants for the different candidate genes showed significant reductions in the number of ENs along the gut for all genes tested, when compared with controls, except for $nos1$ and $pbx3b$ (Fig 2F). In addition to the gut, we imaged whole crispants to visualize additional defects in the whole larval fish body; however, we didn't detect any drastic morphological changes (S2 Fig). Overall, our phenotypic results demonstrate that most CRISPR-sscreened genes are important for ENS establishment, suggesting they may be regulators of ENS formation in zebrafish.

## Genes from the ENS CRISPR screen are expressed along the gut during enteric neurogenesis stages

To assay the expression patterns of ENS candidate genes, we performed Hybridization Chain Reaction (HCR) [42] with probes specific for select gene transcripts in Tg(-8.3$phox2bb$:Kaede) larvae at 96 hpf, such as $ret$, $etv1$, $oprl1$ and $oprd1b$ (Fig 3). In wholemount, we observed specific expression patterning for $ret$, $oprl1$, $etv1$, and $oprd1b$, as well as $elavl3$, in different regions of the brain and/or the spinal cord (S3A–S3F Fig). Gene expression in the developing brain has been consistently reported for genes such as $gfra1a$ and $elavl3$ [27, 43]. Previously, we observed the expression of $oprl1$ within enteric neurons at 70 hpf [20]. As expected, $ret$, $etv1$, $oprl1$ and $oprd1b$ were present along the gut ENs (Fig 3A, 3C, 3E and 3G), with colocalization seen among the Kaede labeled cells (Fig 3B-B", 3D-D", 3F-F" and 3H-H").

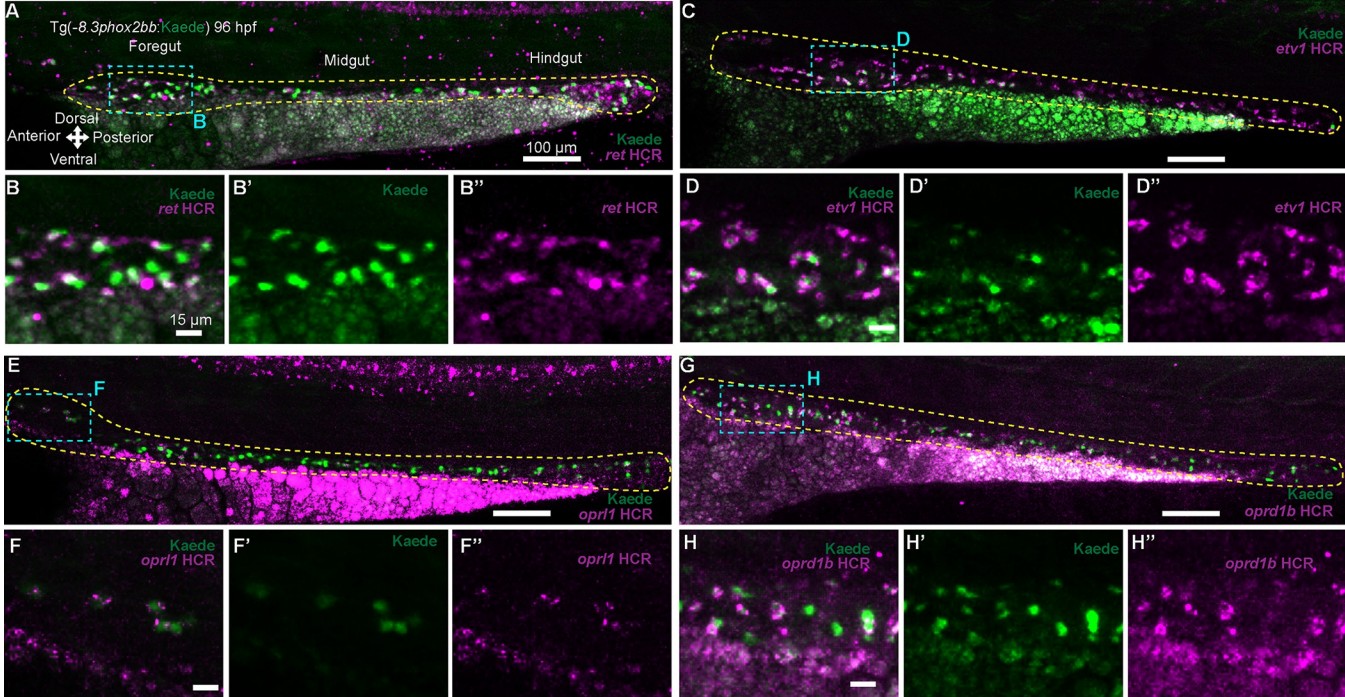

**Fig 3. Expression pattern of CRISPR screen selected genes along the ENS during development.** (A, C, E, G) Confocal images show HCR-assayed expression for *ret*, *etv1*, *oprl1* and *oprd1b* through the gut of Tg(-8.3*phox2bb*:Kaede) larvae at 96 hpf, dashed yellow lines surround the gut. (B-B", D-D", F-F", H-H") Magnified regions of the foregut showing colocalization of *ret*, *etv1*, *oprl1* and *oprd1b* (magenta) with ENCCs expressing the Kaede protein (green). ≥ 3 experiments with 3 biological replicates.

## Chemical inhibition corroborates opioid pathway involvement during ENS development

The opioid receptors have been extensively studied for their physiological roles and as a pharmacological mechanism for pain treatment in the adult GI tract [44, 45]. Opioid receptors have not been implicated in ENS development to date. To further investigate our screen hits, we focused our efforts on chemically targeting the protein products of the opioid receptor encoding genes, the nociception receptor (NOP) Oprl1 (opiate receptor-like 1) or the delta opioid receptor (DOR) Oprd1 (opioid receptor, delta 1a), as crispants for these genes exhibited severe ENS loss in our screen (Fig 2), suggesting functional roles for the opioid pathway during ENS establishment. To that end, we employed pharmacological assays using different opioid inhibitors coupled with ENS differentiation assays in -8.3*phox2bb*:Kaede embryos, starting the treatment at 48 hpf for a duration of 48 hours (Fig 4A). To inhibit Oprl1 or Oprd1, we treated embryos with different antagonists: LY2940094 and curcumin target Oprl1, and the synthetic peptide agonist DADLE, targets Oprd1b [46–48]: From all treated conditions, when compared with DMSO-treated controls, the incubated larvae displayed hypoganglionosis (Fig 4B–4E). Cell counts confirmed the difference in cell number reduction between DMSO and the opioid inhibitors (Fig 4F). These data corroborate a role for the opioid receptors during ENS development. Combined with crispant phenotypic data for *oprl1* and *oprd1b* (Fig 2), these results suggest that the opioid pathway is required for ENS establishment.

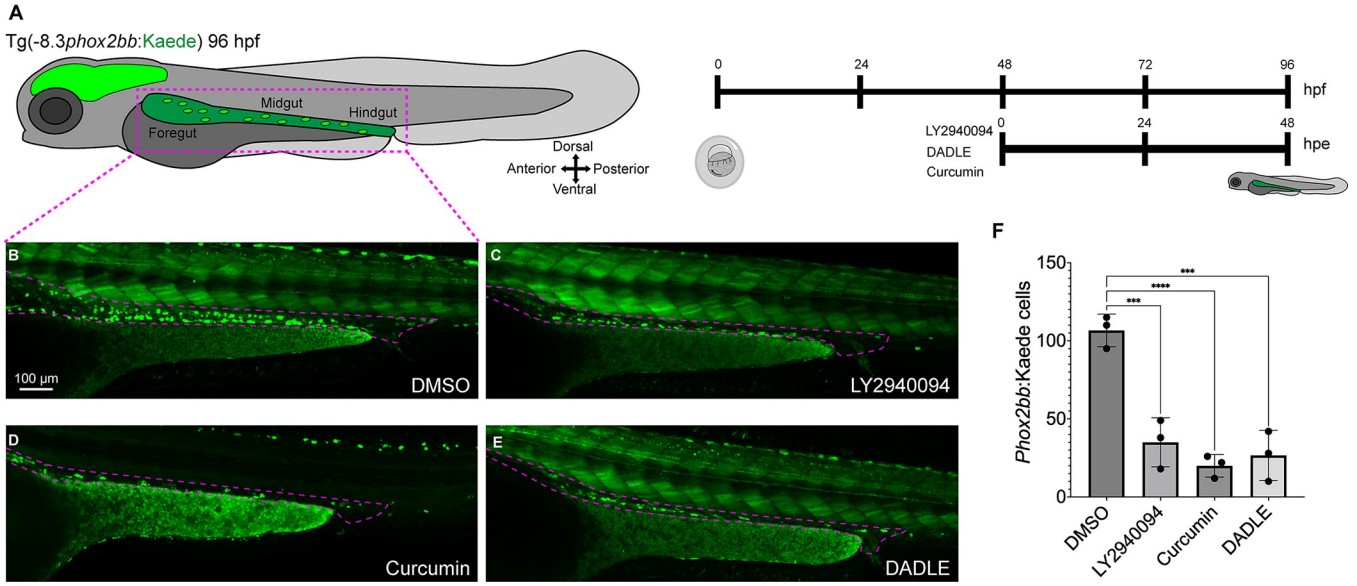

**Fig 4. Temporal chemical inhibition of opioid receptors induces ENS developmental defects in zebrafish larvae.** (A) Tg(-8.3*phox2bb*:Kaede) embryos were exposed at 48 hpf for 48 hpe (hours post exposure) with the opioid inhibitors, LY2940094, curcumin and DADLE, all of them at 10 μM. (B-E) Confocal images reveal fluorescent labeled ENCCs/ENs along the gut in larvae that were treated with DMSO, LY2940094, curcumin and DADLE, respectively. Inhibitor-treated larvae show a reduction of Kaede+ cells, compared with the DMSO control (F) Cell counts of Kaede cells via Imaris, ≥ 3 experiments with 3 biological replicates, mean +/- SEM, ANOVA P value ****: < 0.0001, ***: < 0.0004. Dashed purple lines surround the gut. Number of fluorescent ENCCs and/or ENs along the gut from the different gene crispants.

## Opioid gene crispants have neurochemical coding alterations in the ENS during development

Next, we aimed to examine if the *oprd1b* and *oprl1* crispants displayed ENS phenotypes later during enteric neuronal differentiation stages. We performed wholemount immunohistochemistry at 6 dpf to detect if changes were present in the neurochemical coding of the ENS. To achieve this, we used antibodies against Phox2b [49] and different markers that are present in differentiated ENs, such as HuC/D (Elavl3/4), 5-HT (5-hydroxytryptamine), and Chat (acetylcholine) [19]. Control larvae displayed a complete ENS along the gut, and based on the markers we assayed, showed 5 main populations: Phox2b+/HuC/D+; Phox2b+/HuC/D+/5-HT+; HuC/D+/5-HT+/Chat+; Phox2b+; and HuC/D+ (Fig 5A). Interestingly, while *oprd1b* and *oprl1* crispants generally displayed hypoganglionosis based on the -8.3*phox2bb*:Kaede+ cells, when compared with control, they both had a predominant presence of HuC/D+ ENs, and a relatively smaller number of HuC/D+/5-HT+/Chat+ and Phox2b+/HuC/D+ ENs (Fig 5C and 5E). Despite the altered neurochemical code, these ENs could populate the hindgut. Overall, these imaging data indicate that, when compared with control, the *oprd1b* and *oprl1* crispants display severe ENS neurochemical coding alterations, suggesting that the opioid pathway regulates the proper establishment of ENS during development.

## Discussion

The robust synergy between scRNA-seq data and F0 CRISPR-based reverse genetics in the zebrafish model has allowed us to identify various novel genes involved in establishment of the ENS during development. The selection process for the candidate genes was a concerted effort of bioinformatic analyses and literature curation based on neuronal functions (Table 1). Many

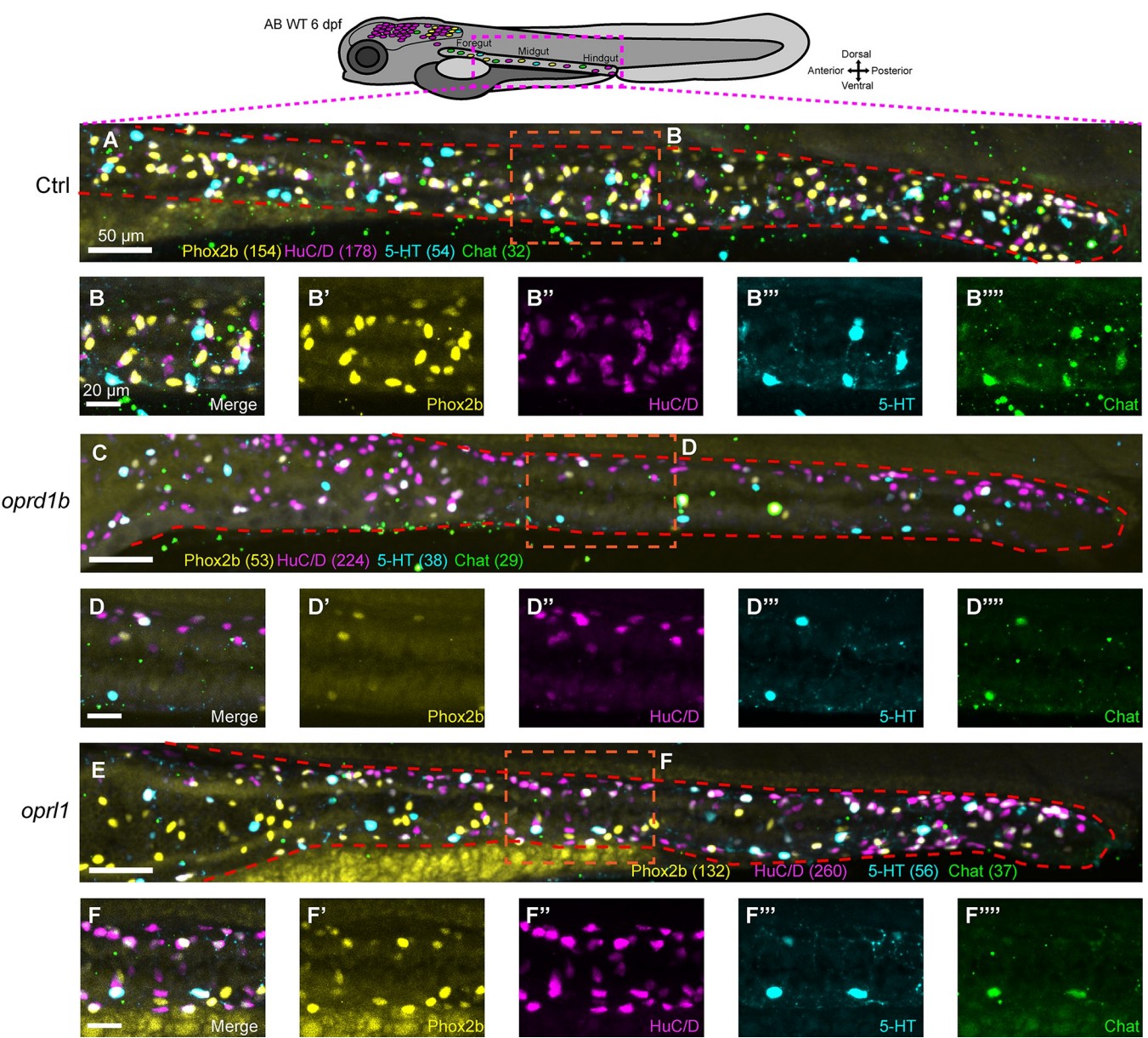

**Fig 5. ENS neurochemical coding is altered in larval crispants for opioid receptor-encoding genes *oprd1b* and *oprl1*.** (A, C, E) Confocal images show whole ENS after immunohistochemistry in 6 dpf control, and *oprd1b* and *oprl1* crispants, respectively. The targeted proteins were Phox2b (yellow), HuC/D (magenta), 5-HT (cyan) and Chat (green). Dashed red lines surround the gut. (B-B"", D-D"", F-F"") depict individual channels and magnification of the ENS midgut region to show the different marker proteins dissected by colors. In parenthesis next to each marker: Cell counts using Imaris software, n = 3 biological replicates.

genes identified from the scRNA-seq differential expression analysis (S1 Data) and used for Metascape and STRING analysis pipelines (S1 Fig) have neuronal functions, and we hypothesized that some more of them may be necessary during ENS development requiring further analysis with our CRISPR targeted screen. The subsequent steps in the screen necessitating indel validations and culminating in high-throughput confocal imaging and analysis proved notably swift, particularly when targeting an ENS colonization phenotype for identification. Overall, we focused on twelve candidate genes, two of which were positive controls already

known to be required for zebrafish ENS development (*ret* and *gfra1a*). Among novel targets, we found that the genes *oprd1b*, *oprl1*, *rufy3*, *etv1*, *vipb*, *vgf*, *ache* and *flot1a*, when mutated, caused significant ENS loss in crispants at 72 hpf (summarized in Table 1). Focusing on *oprd1b* and *oprl1*, we determined that inhibition of the opioid receptors encoded by these genes phenocopied their corresponding crispants, bringing to light the opioid pathway as a regulator of ENS formation.

Our screening approach resembles the study conducted by Gui et al. 2017 [22], wherein they detected *de novo* mutations through exome sequencing of HSCR patients. Similar to our methodology, they utilized the Tg(*-8.3phox2bb*:Kaede) transgenic zebrafish [18] and performed a comparative analysis between morpholino-mediated knockdown and CRISPR knockouts of six genes. Another recent ENS F0 screen identified the role of ten transcription factors, finding alterations in the number of ENs and gut motility using Tg(*phox2bb*:GFP) embryos [50]. A few limitations have been known to occur and vary widely in the generation of crispants models, such as phenotype penetrance, mutagenesis efficiencies, and somatic mosaicism [51]. In our CRISPR ENS screen, we used as a gold standard the *ret* gene to replicate HSCR phenotypes and to assess the efficiency of phenotypes; our *ret* crispant was able to phenocopy the total aganglionosis phenotype (*ret^{wmr1/wmr1}*) [16] with over high 95% efficiency, suggesting that the screen is capable of producing mutations with high penetrance. One of the features of our screen was the capability of identifying most of our candidate genes with alterations in the ENS. This accomplishment was likely due to streamlined analysis of the scRNA-seq sub-clustering and differential expression in combination with Metascape and STRING analyses, demonstrating the ability to rival and produce equivalent outcomes compared to the more intricate approaches [52], where they targeted 188 genes to identify 16 genes that are important for the zebrafish embryonic heart.

Of the twelve candidate genes we screened, phenotyped (Fig 2, Table 1), and validated (Fig 3), two served as positive controls, having known knockout/knockdown zebrafish models. For *ret^{wmr1}* and *ret^{hu2486}* mutants, they present with HSCR-like phenotypes in larval fish [16, 41], and for *gfra1α* morphants, they had a reduction in the number of ENs, displaying hypoganglionosis [27]. To our knowledge, for the additional ten genes, there are no mutational models or phenotypes described in the ENS, highlighting the importance of our screen in illuminating these novel genes with different functions.

Overall, most of our targeted genes have at least one known function related to neurons (S1 Fig and Table 1) and had hypoganglionosis alterations in our screen (Fig 2D). Only *pbx3b* and *nos1* didn't show clear alteration in ENS colonization. For the case of *pbx3b*, this could possibly be explained by a compensation effect, where *pbx1a* may compensate for the loss of *pbx3b*. We found that the gene *pbx1a* is also differentially expressed within the enteric neuronal populations of our scRNA-seq (S1 Data). This possible type of Pbx redundancy has been reported in *Pbx2* null mice, where *Pbx1b* may functionally replace *Pbx2* [53]. Identifying paralogs or gene duplications within our screen with compensation effect will require a multiplex ENS screening that we can foresee for the future. As well, *pbx3b* and *nos1* are expressed almost exclusively in sub-cluster 3 (Fig 1B) suggesting important functions for these two genes when neurons are maturing and/or differentiated [20]. Interestingly, the opioid receptor-encoding genes, *oprd1b* and *oprl1*, and the neuropeptide-encoding genes, *vgf* and *vipb*, are part of the "neuroactive ligand-receptor interaction" KEGG pathway (-log (P value -3.558)), suggesting that during development the ENCCs and the ENS are actively interacting and communicating with other intestinal cell types to mediate diverse functions. This pathway has been found to be upregulated in Parkinson's disease (PD) neurons dependent on miRNAs such as *mir-137* [54]. Looking to the future, it will be important to functionally validate each of our candidate screen

hits to unravel their specific roles during ENS development, and to determine how they may fit within an enteric gene regulatory network.

Of note, the *oprd1b* and *oprl1* crispants and opioid receptor inhibition were sufficient to cause hypoganglionosis in our study, bringing to light the opioid pathway during ENS development. Moreover, when we assayed for changes in the neurochemical coding of ENs in 6-day-old larvae, we found a drastic change in neuronal cell population composition, suggesting an important role of opioid receptors during ENS development and neurogenesis. The opioid receptors have been significantly studied in adult ENS, with respect to the effect of opiates, synthetic opiates and endogenous opioid release, to understanding GI functions, such as motility and secretion [44, 45]. In the mouse brain during development, it has been noted that exposure to morphine and the μ-opioid peptide receptor (MOR) agonist inhibits neural stem and progenitor cells (NSPCs) proliferation by slowing the cell cycle G2/M phase [55]. It is possible a similar process in the developing ENS exists, where ENCCs may slow their proliferation along the gut following downregulation of their opioid receptors (Fig 2D), or by inhibition of them (Fig 4B). Furthermore, in NSPCs, it has been reported that different opioid receptors play a role in neural differentiation and that endogenous opioid systems modulate neural growth and development [56]. Thus, the neurochemical phenotypes we observed in the opioid crispants, where the predominant cell population observed was HuC/D$^+$ (Fig 5), may be due to a general compensatory increase in neural differentiation, or it may signify an alteration in EN subtype differentiation and/or distribution along the gut. This effect was more noticeable with the *oprl1* crispants suggesting additional roles between these opioid receptors besides the initial establishment of ENS.

Ultimately, our zebrafish F0 CRISPR screen of the ENS was able to identify novel genes that are important during ENS development. This screen proved to be efficient by using a combination of scRNA-seq analysis, reverse CRISPR genetics and high-content imaging. In the future, we can envision additional, in-depth functional analysis of each candidate gene, thereby increasing our understanding of ENS development in vertebrates in normal and diseased states.

## Materials and methods

### Zebrafish husbandry, and embryo larvae collection

This work was conducted in accordance with the Institutional Animal Care and Use Committee (IACUC) of Rice University. Embryos and larvae for all experiments were collected from controlled breeding of adult zebrafish for synchronous staging. All embryos were maintained at 28˚C in standard E3 embryo medium until 24 h post fertilization (hpf), then were transferred to 0.003% 1-phenyl 2-thiourea (PTU)/E3 solution [57]. Transgenic embryos used for this work include Tg(−8.3*phox2bb*:Kaede) [18]. Embryos and larvae were collected out of their chorions at the stage noted in each experiment.

### Transcriptomics and enrichment analyses

Expression analysis was done using a zebrafish *sox10*:GFP single-cell RNA-seq atlas, Gene Expression Omnibus (GEO) database accession number GSE152906, and available on UCSC Cell Browser https://cells.ucsc.edu/?ds=zebrafish-neural-crest-atlas [20]. Seurat was used to subset and generate sub-clusters from the 68–70 hpf neural neuronal groups 5 and 12. Data was visualized in feature plots, dot plots or violin plots. Differentially expressed gene markers from sub-cluster 3 (S1 Data) were acquired by using the FindAllMarkers function [20]. Metascape custom enrichment analysis [24] of the sub-cluster 3 differentially expressed gene markers was done by selecting the top 800 genes using gene prioritization by evidence counting

(GPEC) at 0.05 P-value cutoff using the Reactome gene sets from zebrafish. Networks for sub-cluster 3 were generated using STRING and Cytoscape [25, 58].

## CRISPR-Cas9 guide RNA design and synthesis

Twelve sgRNAs were designed using the CRISPR design tool from Synthego (https://design.synthego.com/) using *Danio rerio* (GRCz11) genome and selecting top rank sgRNAs with high activity and minimal off targets [59] (Supplementary Table 2). Negative Control, Scrambled sgRNA #1, GCACUACCAGAGCUAACUCA (Synthego) was used as a negative control in pilot experiments.

## CRISPR-Cas9 microinjections and genotyping

Pools of 30 to 50 embryos fertilized from in-crossing Tg(−8.3*phox2bb*:Kaede) adults were injected at the one-cell stage in the yolk with a solution containing 100 picogram (pg) of gene specific sgRNA, 2 μM Cas9 NLS nuclease (Synthego) and Phenol red. A total of eight injected F0 larvae were dissociated, used in T7 endonuclease I activity assays (NEB E3321) as previously described [16], and the percentage of indels was determined. PCR pair of primers per gene used for the T7E1 activity (S1 Table) targeted the specific sgRNA region for each gene and amplified regions between 200 to 300 bp. Experimental replicates were done at least 3 times per gene for the injections and the genotyping. Live embryos at 48 hpf were genotyped using the Zebrafish Embryonic Genotyper (ZEG) as instructed by the manufacturer (InVivo Biosystems).

## *in vivo* high-content semi-automated confocal microscopy

For the screen, 4 to 8 Tg(−8.3*phox2bb*:Kaede) 72 hpf F0 larvae were selected from each gene specific T7EI confirmed pool, and placed upon a 1% agarose cast made inside of a μ-Dish 35 mm, glass bottom dish (ibidi, 81158). This cast with space for 42 embryos was created from a 3D-printed stamp using a Formlabs Form1+ SLA printer [60]. The embryos were anesthetized using 0.4% Tricaine and covered with a solution of 0.5% low melt temperature agarose dissolved in E3 media. Embedded fish were then covered in 1× PTU/E3 media supplemented with 0.4% Tricaine. Afterwards, confocal imaging was performed in an Olympus FV3000 confocal and FluoView software (2.4.1.198), using an Olympus 10.0X objective (UPLXAPO10X) at a constant temperature of 28˚C, maintained with an OKOLAB Uno-controller imaging incubator. The embedded fish were scanned in an automated fashion using the multi-area time-lapse software module (MATL). Z-stack images of the ENS were combined using the Fiji Image-J stitch plugin version 1.2 and then processed and exported in IMARIS image analysis software (Bitplane) to quantify cell numbers. Figures were prepared in Adobe Illustrator software.

## Hybridization chain reaction and whole mount immunohistochemistry

Hybridization chain reaction (HCR) and Whole Mount Immunohistochemistry were done in accordance with previous described methods [42, 61]. The HCR probes transcripts were synthesized by Molecular Instruments for *ret*, NM_181662.2; *gfra1a*, NM_131730.1; *oprl1*, NM_205589.2; *oprd1b*, *NM_131258.4*; *etv1*, XM_005157634.4; *elavl3*, NM_131449. The following primary antibodies were used: goat polyclonal IgG anti-Choline Acetyltransferase (ChaT, Millipore Sigma, AB144P, 1:500), rabbit polyclonal IgG anti-5-HT (serotonin, Immunostar, 20080, 1:250), mouse monoclonal IgG2b anti-HuC/D (Invitrogen Thermo Fisher, A-21271, 1:250), Mouse monoclonal IgG1 anti-Phox2b (B-11, Santa Cruz Biotechnology,

SC-376997, 1:250). The following secondary antibodies were used from Invitrogen: Alexa Fluor 488 donkey anti-goat IgG (A-11055, 1:600), Alexa Fluor 405 goat anti-rabbit IgG (A-48254, 1:600), Alexa Fluor 647 goat anti-mouse IgG2b (A-21242, 1:600), Alexa Fluor 594 goat anti-mouse IgG1 (A-21125, 1:600). High content semi-automated confocal imaging and processing was done as described above.

## Zebrafish treatment with chemical inhibitors

LY2940094 (MedChemExpress, HY-114452), curcumin (sigma-aldrich, C1386) and DADLE (abcam, ab120673) master stocks were diluted in DMSO and then further diluted in 1xPTU/ E3 medium to the required working concentration (10 μM). Eight Tg(−8.3phox2bb:Kaede) embryos per well were set up on a 24-well plate (Corning, CLS3527) with 1 ml of 1xPTU/E3. Drug was added at 48 hpf and incubated for 48 hpe (hours post exposure, until 96 hpf). Titration experiments were done from low (5μM), medium (10μM), and high concentrations (50 μM) to determine the 10 μM working concentrations. 48 hpf was chosen, as this is when ENCCs are actively migrating along the gut to colonize and undergo neurogenesis [62, 63]. Larvae were extensively washed to remove treatments, then prepared and imaged by high-content semi-automated confocal microscopy as described above. Experimental replicates were done 3 times with over 4 biological replicates each.

## Statistics

Statistical analyses were performed in GraphPad Prism (version 10.1.1). For comparisons, data was tested using a two-tailed unpaired t-test and Ordinary one-way ANOVA, *$P<0.05$, n.s., non-significant ($P<0.05$).

## Supporting information

**S1 Fig. Selection of ENS genes for the CRISPR screen was based on scRNA seq analysis, Metascape functional enrichment, and STRING analysis.** (A) Dot plot depicts the expression level of enteric neuron-specific markers across individual clusters generated within the main 68–70 hpf tSNE sox10:GFP+ dataset [20]. Clusters 5 and 12 prominently expressed enteric neuron markers. Dot size depicts the cell percentage for each marker, and the color summarizes the average expression levels for each gene. (B, C) Metascape network of enriched terms from the Reactome zebrafish gene set, colored by cluster (B) or by p-values (C), this network was based on the differentially expressed genes table from Seurat scRNA seq analysis of sub-cluster 3 (S1 Data). (D) Bar graph of top 20 enriched Reactome zebrafish terms across input gene lists colored by p-values. (E) STRING network based on the complete sub-cluster 3 depicting the selected genes for the CRISPR screen in red.
(TIF)

**S2 Fig. Whole-body crispant phenotyping did not identify drastic morphological defects.** Confocal images of whole Tg(-8.3phox2bb:Kaede) crispants for ret (B), oprd1b (C), oprl1 (D), and control (A) at the 72 hpf, with no visible drastic effects ($\geq$ 3 experiments with 3 biological replicates).
(TIF)

**S3 Fig. Whole-body crispant phenotyping and expression patterns of selected CRISPR screen ENS genes.** Confocal images show HCR-assayed expression of ret (A), etv1 (B), oprl1 (C), oprd1b (D), gfra1a (E), and elavl3 (F) in whole Tg(-8.3phox2bb:Kaede) larvae at the 96 hpf, revealing expression along the spinal cord and brain regions. The Kaede signal is green, and the specific mRNA signal is magenta. White arrows depict the signals of the probes in the

brain or spinal cord. Dashed yellow lines surround the ENS.
(TIF)

**S1 Data. Differentially expressed genes of sub-cluster 3 from Seurat scRNA seq [20].**
(XLSX)

**S1 Table. sgRNAs that target the different ENS candidate genes.**
(XLSX)

## Acknowledgments

We want to express our sincere appreciation to Margarita Niño, Lucia J. Rivas, and James J. Tallman, for their invaluable assistance in initiating experiments. Also, we want to thank Aubrey GA Howard IV, Phillip A Baker, and Helen Folasade Adu for their insight, advice, and technical assistance.

## Author Contributions

**Conceptualization:** Rodrigo Moreno-Campos, Rosa A. Uribe.

**Data curation:** Rodrigo Moreno-Campos.

**Formal analysis:** Rodrigo Moreno-Campos, Eileen W. Singleton, Rosa A. Uribe.

**Funding acquisition:** Rosa A. Uribe.

**Investigation:** Rodrigo Moreno-Campos, Eileen W. Singleton.

**Methodology:** Rodrigo Moreno-Campos, Rosa A. Uribe.

**Project administration:** Rodrigo Moreno-Campos, Rosa A. Uribe.

**Resources:** Rosa A. Uribe.

**Supervision:** Rosa A. Uribe.

**Validation:** Rodrigo Moreno-Campos, Eileen W. Singleton.

**Visualization:** Rodrigo Moreno-Campos, Rosa A. Uribe.

**Writing – original draft:** Rodrigo Moreno-Campos.

**Writing – review & editing:** Rodrigo Moreno-Campos, Rosa A. Uribe.

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
