## [Decision Letter · Decision Letter 0]

5 Mar 2024

PONE-D-24-02185A targeted CRISPR-Cas9 mediated F0 screen identifies genes involved in establishment of the enteric nervous systemPLOS ONE

Dear Dr. Uribe,

Thank you for submitting your manuscript to PLOS ONE. After careful consideration, we feel that it has merit but does not fully meet PLOS ONE’s publication criteria as it currently stands. Therefore, we invite you to submit a revised version of the manuscript that addresses the concerns raised by the reviewers.

We look forward to receiving your revised manuscript.

Kind regards,

Yevgenya Grinblat, Ph. D.

Academic Editor

PLOS ONE

“None”

4. We note that Figure 1 in your submission contain copyrighted images. All PLOS content is published under the Creative Commons Attribution License (CC BY 4.0), which means that the manuscript, images, and Supporting Information files will be freely available online, and any third party is permitted to access, download, copy, distribute, and use these materials in any way, even commercially, with proper attribution. For more information, see our copyright guidelines: http://journals.plos.org/plosone/s/licenses-and-copyright.

Reviewers' comments:

Reviewer's Responses to Questions

**Comments to the Author**

1. Is the manuscript technically sound, and do the data support the conclusions?

Reviewer #1: Yes

Reviewer #2: Yes

2. Has the statistical analysis been performed appropriately and rigorously? 

Reviewer #1: Yes

Reviewer #2: Yes

3. Have the authors made all data underlying the findings in their manuscript fully available?

Reviewer #1: Yes

Reviewer #2: Yes

4. Is the manuscript presented in an intelligible fashion and written in standard English?

Reviewer #1: Yes

Reviewer #2: Yes

5. Review Comments to the Author

Reviewer #1: The manuscript by Moreno-Campos et al uses the zebrafish to examine gene function by completing a Crispr F0 screen for factors identified as enriched in enteric neural crest cells (ENCC). Specifically, the authors tested the idea that this approach can be used as a pipeline to identify the function of novel genes. They utilize imaging approaches to identify the phenotypes including a novel role for opioid receptors in ENCC devlopment. The manuscript is well written and illustrated, includes a nice table of the gene function and phenotype observed and is of interest to the readers of PlosOne. There are a few suggestions that would strengthen the manuscript. My comments are below:

1. It would be interesting to explain a bit more how these specific targets were chosen for Crispr knockdown. The authors define them as high cell expressed distribution but this list I imagine is large. And from the STRING analysis it is not clear that they cluster in similar ways. Please clarify this aspect in the text.

2. In Figure 2 it shows the pipeline for the analysis. Was each imaged embryo shown in panel D genotyped? With this high of efficiency, it is probably not necessary but useful to confirm that the embryos that you are observing have cuts in those embryos.

3. For consistency, quantification should be completed on the opioid inhibitors and crispants as well in terms of the total number of cells in each condition is needed. It is also curious that oprl1 is less effective the oprd1b. What are the authors thoughts on this?

4. It would be useful to add a discussion of the pros and cons of this approach. Because of the nature of the experiment the embryos will have mosaic knockdown and thus may not show the actual phenotype in all cells. How do the authors think this effects the results presented? This approach also does not address the genetic redundancy issue that a germ line mutant may have and thus the need to knockout multiple paralogs to see a similar phenotype to those shown here.

Minor:

1. In some of the images in Figure 2, there is gfp expression in the somites, is that just leakiness of the transgene?

2. Also for quantification of the phox2bb positive cells it would be more rigorous in my opinion to show the total number of cells in each to do the statistics on instead of the percentage of cells.

Reviewer #2: The manuscript by Moreno-Campos et al describes results of rapid functional characterization of several candidate genes with suspected roles in the developing enteric nervous system. The candidates were chosen based on their expression in the enteric neuronal progenitors, and F0 CRISPR/Cas9 mutagenesis was used to generate mosaic mutant embryos (crispants) in each of these genes. The approach appears to be robust and has identified a novel role for opioid receptors in the developing ENS. However, there are several deficiencies that preclude recommending publication in its current form. The manuscript will be of interest to a broad audience of zebrafish geneticists and beyond after these concerns are addressed.

Since the candidate genes are not restricted to the ENs, it is important to consider the effects of mutagenesis on the entire embryo in order to evaluate specificity of the observed EN defects. How was crispant morphology assessed prior to EN imaging? Did all crispants develop normally outside of the ENS?

Figure 1 is too cluttered to be helpful in its current form. Graphics representing bioinformatic analysis, CRISPR/Cas9, the adult fish and confocal microscope can all be removed. It would be helpful if the arrows that indicate the order of steps zigzagged less.

Figures 4 and 5: how many embryos were imaged, and in how many independent experiments? How were the numbers of GFP cells quantified? What were these numbers? Some of this information is in the Methods section but it needs to be included explicitly in the figure legends as well.

Overall, the Results section is unnecessarily wordy and at times inappropriately flowery. For example, the sentence “Our efforts to locate literature pertaining to opioid receptor involvement in ENS development proved elusive” can be written more concisely as “opioid receptors have not been implicated in ENS development to date” without loss of information. At the same time, this section lacks important details about how the results were obtained. Some of these details are contained in the figure legends and should be moved to the Results.

Discussion is overly focused on the advantages of the screening methodology and does not do enough to relate these findings to the broader field and to define specific next steps.

There are numerous grammatical and punctuation mistakes throughout the manuscript. Rigorous editing is required prior to resubmission for the manuscript to meet publication standards for the journal.

6. PLOS authors have the option to publish the peer review history of their article (what does this mean?). If published, this will include your full peer review and any attached files.

Reviewer #1: No

Reviewer #2: No

---

## [Author Response · Author response to Decision Letter 0]

20 Apr 2024

Response to Reviewers and Editor

Title:

A targeted CRISPR-Cas9 mediated F0 screen identifies genes involved in establishment of the enteric nervous system 

PONE-D-24-02185

We are thankful for PLOS ONE’s interest and careful consideration of our work. We especially thank reviewers for their constructive suggestions. We wish to share our point-by-point revisions and responses for addressing the referee’s comments made for resubmission.

Per Reviewer #1 Comments:

1. “It would be interesting to explain a bit more how these specific targets were chosen for Crispr knockdown. The authors define them as high cell expressed distribution but this list I imagine is large. And from the STRING analysis it is not clear that they cluster in similar ways. Please clarify this aspect in the text”.

We have added a more detailed explanation of how these genes were chosen in the discussion: “The selection process for the candidate genes was a concerted effort of bioinformatic analyses and literature curation based on neuronal functions (Table 1). Many genes identified from the scRNA-seq differential expression analysis (S1 Data) and used for Metascape and STRING analysis pipelines (S1 Fig) have neuronal functions, and we hypothesized that some more of them may be necessary during ENS development requiring further analysis with our CRISPR targeted screen.” Related to that, we updated Table 1 headers to reflect that the information shown includes zebrafish gene symbols, gene names, and curated information from the literature about candidate gene function.

2. In Figure 2 it shows the pipeline for the analysis. Was each imaged embryo shown in panel D genotyped? With this high of efficiency, it is probably not necessary but useful to confirm that the embryos that you are observing have cuts in those embryos.

For the genotyping pipeline, a representative number of embryos (n=8) was selected for the 48 hpf CRISPR pool (n=30). These representative embryos with over 95% indel detection ensure that the probability of the other ~22 remaining embryos will also have indels. As kindly suggested, we have included information regarding live genotyping before confocal imaging to support the high indel activity of crispants in the final paragraph of the “Candidate gene crispants have ENS genotypic and phenotypic alterations” results section. We also modified Fig 2B by adding a representative zebrafish embryonic genotyper (ZEG) T7E1 assay of the etv1 gene and the figure captions accordingly.

3. For consistency, quantification should be completed on the opioid inhibitors and crispants as well in terms of the total number of cells in each condition is needed. It is also curious that oprl1 is less effective the oprd1b. What are the authors thoughts on this?

Cell counts were completed on the opioid inhibitor and crispant experiments. Both experiments now include total phox2bb:Kaede+ cells (Fig 2F, Fig 4F). For opioid inhibitor experiments, a new panel was added in Fig 4F with the cell counts. The corresponding text was added in the “Chemical inhibition corroborates opioid pathway involvement during ENS development” results section. We didn’t have a statistical difference in total enteric cell numbers between oprl1 and oprd1b crispants. However, we have seen differences between the neurochemical code and the HuC/D+ marker (Fig 5). We appreciate the reviewer bringing this to our attention and have added this difference between the oprl1 and oprd1b receptors in the discussion.

4. It would be useful to add a discussion of the pros and cons of this approach. Because of the nature of the experiment the embryos will have mosaic knockdown and thus may not show the actual phenotype in all cells. How do the authors think this effects the results presented? This approach also does not address the genetic redundancy issue that a germ line mutant may have and thus the need to knockout multiple paralogs to see a similar phenotype to those shown here.

We have added in the discussion the limitations of F0s screens, such as variations in phenotype penetrance, mutagenesis efficiencies, and somatic mosaicism. Additionally, we demonstrate that the ret crispants we generated and used as an ENS phenotypic positive control could replicate a complete aganglionosis phenotype that happens in biallelic edited F2 families. We also discussed the capabilities of our screen to do multiplex editions to limit the compensation effect of paralogues and gene duplication.

For the minor suggestions:

1. In some of the images in Figure 2, there is gfp expression in the somites, is that just leakiness of the transgene?

Yes, we have sometimes seen leakiness of the transgene in the somites. It is not a specific phenomenon based on endogenous phox2bb gene expression patterns. Nonetheless, the ENS pattern is robust from clutch to clutch.

2. Also for quantification of the phox2bb positive cells it would be more rigorous in my opinion to show the total number of cells in each to do the statistics on instead of the percentage of cells.

We have changed quantifications to positive cells to be more rigorous than the percentage of cells.

Per Reviewer 2 Comments:

1. Since the candidate genes are not restricted to the ENs, it is important to consider the effects of mutagenesis on the entire embryo in order to evaluate specificity of the observed EN defects. How was crispant morphology assessed prior to EN imaging? Did all crispants develop normally outside of the ENS?

To consider the mutagenesis effects of whole embryo, in the “Candidate gene crispants have ENS genotypic and phenotypic alterations” results section, we added a sentence mentioning the lack of global morphological phenotypes; we also added a new supplementary figure showing representative ret, oprd1b and oprl1 crispants compared with a control embryo as suggested (S2 Fig). We look for common zebrafish phenotypic visible defects such as fin shape, heart size, and phox2bb expression in the brain. We were not able to detect any drastic visible morphological defects.

2. Figure 1 is too cluttered to be helpful in its current form. Graphics representing bioinformatic analysis, CRISPR/Cas9, the adult fish and confocal microscope can all be removed. It would be helpful if the arrows that indicate the order of steps zigzagged less.

For Fig 1, we have changed the angle of the arrows to avoid zigzagging, substituted the confocal microscope, and simplified the illustrations. Besides that, we respectfully disagree; we think that by removing our artwork, this figure will appeal to fewer people in search engines. These illustrations aim to reach and generate interest from people outside our field. In addition, it works as a very general figure that can explain our screen and be understood in a very small amount of time. Finally, the artwork of Fig 1 is followed by all the other figures to improve the understanding for more visual scientists. We want to follow the trend of scientific literature that is more visual and appealing.

3. Figures 4 and 5: how many embryos were imaged, and in how many independent experiments? How were the numbers of GFP cells quantified? What were these numbers? Some of this information is in the Methods section but it needs to be included explicitly in the figure legends as well.

We have added the number of embryos imaged and the number of experiments in the figure’s legends. For example, in Fig 2F, “Number of fluorescent ENCCs and/or ENs along the gut from the different gene crispants (≥ 3 experiments with 3 biological replicates). Comparing the mean of the control with the mean of each gene, ANOVA P value ****: <0.0001, ***: 0.0004, **: <0.003, *: <0.05.” As described in the Methods, the number of GFP cells was quantified using IMARIS software.

4. Overall, the Results section is unnecessarily wordy and at times inappropriately flowery. For example, the sentence “Our efforts to locate literature pertaining to opioid receptor involvement in ENS development proved elusive” can be written more concisely as “opioid receptors have not been implicated in ENS development to date” without loss of information. At the same time, this section lacks important details about how the results were obtained. Some of these details are contained in the figure legends and should be moved to the Results.

We found unnecessary text and substituted it with a concise text, as suggested, throughout the results and discussion. We have checked to ensure all information about how results were obtained is clear in the figures, results, and methods. We also added more findings from the discovery biology to the discussion. We discussed how oprl1, oprd1b, vgf, and vipb encode proteins that are part of the neuroactive ligand-receptor interaction pathway and how this pathway may regulate, communicate, and interact with other intestinal cells.

5. There are numerous grammatical and punctuation mistakes throughout the manuscript. Rigorous editing is required prior to resubmission for the manuscript to meet publication standards for the journal.

We have run a grammar assistant to correct the punctuation mistakes and made some suggested changes. 

Per Academic Editor Comments:

We have created a marked-up copy highlighting in yellow the changes made to the original version.

Journal Requirement 3. We removed the “data not shown” in paragraph 4 in the discussion section. Instead, we have modified the text by adding in the paragraph the presence of another paralogue, pbx1a, present in our S1 Data. We have also added the citation from Selleri et al., 2004, suggesting a functional redundancy from other pbx isoforms.

In the zebrafish treatment with chemical inhibitors from Materials and Methods, we have changed the text to remove “data not shown.” We describe that the working concentration was based on a selection of concentrations from low (5µM), medium (10µM), and high concentrations (50 µM) to determine the 10 µM working concentration.

Journal Requirement 4. For Figure 1, the confocal microscope cartoon was substituted with one that we created. We also simplify some of the other illustrations. This way, we created all the figures.

5. We have updated the in-text citations accordingly and modified the figure captions. Also, we have moved the supporting information files to the very end of the manuscript.

---

## [Decision Letter · Decision Letter 1]

3 May 2024

A targeted CRISPR-Cas9 mediated F0 screen identifies genes involved in establishment of the enteric nervous system

PONE-D-24-02185R1

Dear Dr. Uribe,

I am happy to share that your revised manuscript has been judged suitable for publication and will be formally accepted for publication once it meets all outstanding technical requirements.

Kind regards,

Yevgenya Grinblat, Ph. D.

Academic Editor

PLOS ONE

Reviewers' comments:

Reviewer's Responses to Questions

**Comments to the Author**

1. If the authors have adequately addressed your comments raised in a previous round of review and you feel that this manuscript is now acceptable for publication, you may indicate that here to bypass the “Comments to the Author” section, enter your conflict of interest statement in the “Confidential to Editor” section, and submit your "Accept" recommendation.

Reviewer #1: All comments have been addressed

Reviewer #2: All comments have been addressed

2. Is the manuscript technically sound, and do the data support the conclusions?

Reviewer #1: Yes

Reviewer #2: Yes

3. Has the statistical analysis been performed appropriately and rigorously? 

Reviewer #1: Yes

Reviewer #2: Yes

4. Have the authors made all data underlying the findings in their manuscript fully available?

Reviewer #1: Yes

Reviewer #2: Yes

5. Is the manuscript presented in an intelligible fashion and written in standard English?

Reviewer #1: Yes

Reviewer #2: Yes

6. Review Comments to the Author

Reviewer #1: The authors have addressed all my concerns. They have nicely updated the figures and statistics and now is a stronger paper.

Reviewer #2: The authors have thoroughly addressed my concerns and have further strengthened this important and rigorous study.

7. PLOS authors have the option to publish the peer review history of their article (what does this mean?). If published, this will include your full peer review and any attached files.

Reviewer #1: No

Reviewer #2: No

---

## [Editor Report · Acceptance letter]

7 May 2024

PONE-D-24-02185R1 

PLOS ONE

Dear Dr. Uribe, 

I'm pleased to inform you that your manuscript has been deemed suitable for publication in PLOS ONE. Congratulations! Your manuscript is now being handed over to our production team.

Kind regards, 

on behalf of

Dr. Yevgenya Grinblat 

Academic Editor

PLOS ONE